# Metabolic and Molecular Basis of Sarcopenia: Implications in the Management of Urothelial Carcinoma

**DOI:** 10.3390/ijms20030760

**Published:** 2019-02-11

**Authors:** Hiroshi Fukushima, Yasuhisa Fujii, Fumitaka Koga

**Affiliations:** 1Department of Urology, Tokyo Medical and Dental University Graduate School, 1-5-45 Yushima, Bunkyo-ku, Tokyo 113-8510, Japan; y-fujii.uro@tmd.ac.jp; 2Department of Urology, Tokyo Metropolitan Cancer and Infectious diseases Center Komagome Hospital, 3-18-22 Honkomagome, Bunkyo-ku, Tokyo 113-8677, Japan

**Keywords:** sarcopenia, biomarker, urothelial carcinoma

## Abstract

Sarcopenia, which represents the degenerative and systemic loss of skeletal muscle mass, is a multifactorial syndrome caused by various clinical conditions. Sarcopenia reflects not only frailty and poor general health status, but also the possible presence of advanced or progressive cancer or cancer cachexia. Therefore, sarcopenia affects the management of cancer-bearing patients, including those with urothelial carcinoma. Recently, growing evidence has shown that sarcopenia is significantly associated with higher rates of treatment-related complications and worse prognosis in patients with urothelial carcinoma, including muscle-invasive bladder cancer, upper tract urothelial carcinoma, and advanced urothelial carcinoma. Moreover, several studies reported that a post-therapeutic increase in skeletal muscle mass predicts favorable prognosis in urothelial carcinoma patients. To further explore the role of sarcopenia in the management of urothelial carcinoma patients, comprehensive understanding of its pathophysiology is vital. In this article, we reviewed the metabolic and molecular basis of cancer cachexia and sarcopenia. From this viewpoint, we discussed the possible mechanism of changes in skeletal muscle mass during the course of treatment.

## 1. Introduction

Urothelial carcinoma, which develops from the urothelium of the renal pelvis, ureter, and bladder, is the most prevalent histological type of malignancy of the urinary tract. It is mainly comprised of bladder cancer and upper tract urothelial carcinoma (UTUC). Bladder cancer accounts for over 90% of urothelial carcinoma, and thus is considered as a common genitourinary malignancy in the United States, with approximately 81,000 new cases and 17,000 deaths each year as of 2018 [1]. Meanwhile, UTUC is a relatively rare malignant disease, with an incidence of two cases per 100,000 person-years in the United States [2]. Bladder cancer is categorized into muscle-invasive bladder cancer (MIBC) and non-muscle-invasive bladder cancer (NMIBC) according to the pathological depth of the tumor invasion. MIBC, which accounts for approximately 25% of all new bladder cancer cases, is related to higher rates of metastasis compared with NMIBC [3]. MIBC patients are generally treated with radical cystectomy and urinary diversion, and nearly half of them recur and eventually die within five years postoperatively, despite undergoing invasive surgery [4]. As for UTUC, over 40% of patients with UTUC already have locally advanced or metastatic disease at the initial treatment [2]. In addition, over 20% of patients with localized UTUC experience metastatic recurrence following radical nephroureterectomy, despite undergoing curative surgery [5]. Although platinum-based chemotherapy, which occasionally causes serious adverse events to patients, is the standard first-line therapy for metastatic urothelial carcinoma, the prognosis is unfavorable, with a median overall survival (OS) of approximately 15 months [6]. Recently, the advent of immuno-oncology drugs has led to a paradigm shift regarding the therapeutic strategies for urothelial carcinoma, but long-term efficacy is observed in only approximately 20% of patients [7]. Given the limited effectiveness and complication risks of the treatments for urothelial carcinoma, risk assessment based on biomarkers is important for clinicians to predict prognosis and complication risk, determine treatment plans, and counsel patients in the management of urothelial carcinoma.

Sarcopenia, which represents the degenerative and systemic loss of skeletal muscle mass, is a multifactorial syndrome caused by aging, physical inactivity, malnutrition, neuromuscular disorders, inflammatory conditions, endocrine diseases, malignancies, and so on [8,9]. Recent surveys showed a high prevalence of sarcopenia, ranging from 15% at 65 years to 50% at 80 years [10]. Sarcopenia is associated with poor physical performance and a higher risk of fall and fracture [11,12]. In addition, sarcopenic patients tend to have higher rates of morbidity from infectious diseases [13], metabolic syndrome [14], insulin resistance [15], and cardiovascular diseases and higher rates of mortality [16]. Thus, sarcopenia reflects frailty and the general health status of patients. Moreover, sarcopenia can represent the presence of cancer cachexia [9]. The metabolic balance of patients with cancer cachexia shifts towards a catabolic state rather than an anabolic state because of anorexia, poor nutrition, and systemic inflammation. This leads to catabolism of skeletal muscle and results in sarcopenia. Therefore, sarcopenia is considered as an indicator of not only poor general health status, but also the possible presence of progressive or advanced cancer.

Recently, a growing body of evidence showed the prognostic significance of sarcopenia in various cancers, including lung or gastrointestinal cancer [17,18], hepatic cell carcinoma [19], esophageal cancer [20], lymphoma [21], melanoma [22], and renal cell carcinoma [23,24]. Moreover, sarcopenia can contribute to higher rates of treatment-related complications in various cancers, including those due to surgical treatment, chemotherapy, or tyrosine kinase inhibitors [25,26,27]. As for urothelial carcinoma, many studies reported that sarcopenia was significantly associated with higher rates of treatment-related complications and worse prognosis [28]. Sarcopenia was a significant predictor for higher rates of perioperative complications and worse cancer-specific survival after radical cystectomy [29,30]. The prognostic significance of sarcopenia was also reported in UTUC patients treated with radical nephroureterectomy [31,32] and in those with advanced urothelial carcinoma [33], which includes inoperable locally advanced and/or metastatic disease. Moreover, the recovery of skeletal muscle mass after chemotherapy was significantly associated with favorable prognosis in advanced urothelial carcinoma patients treated with first-line platinum-based chemotherapy [34]. These findings suggest that sarcopenia can be a clinically relevant syndrome especially in urothelial carcinoma. To further explore the role of sarcopenia in the management of urothelial carcinoma patients, it is necessary to comprehensively understand the pathophysiology of sarcopenia. In this article, we review the metabolic and molecular basis of cancer cachexia and sarcopenia. Moreover, we discuss the possible mechanism of changes in skeletal muscle mass in urothelial carcinoma patients based on the underlying metabolic and molecular basis.

## 2. Metabolic and Molecular Basis of Cancer Cachexia and Sarcopenia

According to the international consensus of cancer cachexia, it is defined as a multifactorial syndrome characterized by an ongoing loss of skeletal muscle mass (with or without loss of fat mass) that cannot be fully reversed by conventional nutritional support, leading to progressive functional impairment [9]. Thus, sarcopenia reflects one of the manifestations of cancer cachexia, but sarcopenia includes skeletal muscle mass loss caused by other etiologies, such as aging and physical inactivity.

Sarcopenia can be induced by various mechanisms in cancer cachexia (Figure 1). Most patients with cancer cachexia have anorexia, which is caused by cancer itself or treatment for cancer, such as systemic chemotherapy, leading to decreased protein synthesis. Hypercatabolism is caused by systemic inflammation, which is mediated by increased secretion of proinflammatory and inflammatory cytokines, resulting in increased protein degradation. These processes can promote hypotrophy of myofibers. In addition, apoptosis of myofibers is caused by increased oxidative stress. Oxidative stress can be caused by excessive fatty acid oxidation, in which free fatty acids are provided by enhanced depletion of adipose tissue in the early phase of cancer cachexia [35]. These mechanisms can be stimulated by various mediators, such as cytokines [36]. In this section, we review the mechanism of body weight loss and the depletion of adipose tissue during cancer cachexia. We also summarize general regulators of skeletal muscle mass homeostasis and the metabolic and molecular background of sarcopenia associated with cancer cachexia.

### 2.1. Energy Balance in Cancer Cachexia

Cancer cachexia is characterized by progressive weight loss, which results from decreased energy intake and increased energy expenditure [9]. The decrease in energy intake can be caused by anorexia. In patients with cancer cachexia, cytokines, such as interleukin (IL)-1, IL-6, and tumor necrosis factor (TNF)-α, are transported across the blood-brain barrier, and induce anorexia by modulating the hypothalamic areas of the brain, the key region for appetite regulation [37]. Moreover, an increase in energy expenditure can occur as the wasting process is characterized by systemic inflammation. In addition, resting energy expenditure gains are observed as a result of increased thermogenesis caused by white adipose tissue browning [38]. This is related to upregulated expression of uncoupling protein 1 (UCP1), which mediates proton leakage across the inner mitochondrial membrane and then provokes uncoupling of respiration to adenosine diphosphate phosphorylation [38]. White adipose tissue browning can be induced by β-adrenergic receptor stimulation and cytokines, such as TNF-α and IL-6 [38]. These mechanisms causing energy imbalance contribute to the depletion of adipose and skeletal muscle tissues in cancer cachexia.

### 2.2. Mechanisms of Adipose Tissue Depletion in Cancer Cachexia

Fatty acids, which are stored in adipose tissue as triacylglycerols, are hydrolyzed from plasma lipoproteins by lipoprotein lipase and transported into adipocytes to synthesize triacylglycerols. Lipolysis is mediated by various hormones, such as epinephrine, glucagon, and adrenocorticotrophic hormone, through a cyclic adenosine monophosphate (cAMP)-mediated process [37]. Hormone-sensitive lipase (HSL), a key enzyme for the conversion of one molecule of triacylglycerols into three molecules of fatty acids and one molecule of glycerol, is activated by a protein kinase, which is activated by cAMP [37]. Adipose triglyceride lipase (ATGL) is another enzyme related to the catabolism of triacylglycerols [37].

Lipolysis is activated in the early phase of cancer cachexia [35]. In cancer cachexia, lipolysis is activated by various factors, including enhanced stimulation of β-adrenergic receptors; increased secretion of cytokines, such as TNF-α, IL-1, IL-6, and interferon-γ; and increased expression of lipid-mobilizing factors, such as zinc-α2 glycoprotein-1 (AZGP1) [37]. Triglyceride lipolysis causes the breakdown of adipose tissue. Previous studies on ATGL- or HSL-deficient animal models showed that the absence of ATGL, and, to a lesser degree, HSL, reduces fatty acid mobilization and ameliorates the depletion of adipose tissue, resulting in the inhibition of skeletal muscle wasting [39]. Thus, excessive lipolysis may progress skeletal muscle wasting.

### 2.3. General Regulators of Skeletal Muscle Mass Homeostasis and Sarcopenia

Sarcopenia results from hypotrophic or apoptotic changes in myofibers from a microscopic viewpoint. In this section, we summarize the general regulators of skeletal muscle mass homeostasis and sarcopenia regardless of the presence of cancer cachexia.

#### 2.3.1. Hypotrophic Changes in Myofibers

Hypotrophic changes in myofibers can be caused by various extrinsic and intrinsic factors. Poor nutritional status can lead to reduced protein synthesis. Persistent systemic inflammation, which is caused by various diseases, such as infection, inflammatory diseases, and cancer, facilitates the degradation of proteins. The key molecules of the signaling pathways in these processes include Akt. Skeletal muscle mass homeostasis is generally regulated by the phosphatidylinositol-3 kinase (PI3K)/Akt pathway, a mammalian target of rapamycin (mTOR), and forkhead box O (FOXO) [37,40]. Insulin-like growth factor 1 (IGF-1) activates PI3K/Akt signaling and eventually increases the expression of Atrogin-1 and the muscle ring finger protein 1 (MuRF-1), muscle-specific E3 ligases that play a key role in regulating ubiquitin-driven protein degradation in skeletal muscle tissues, via the suppression of FOXO [36,40]. In addition, activated PI3K/Akt signaling results in an increase in protein synthesis by activating the mTOR complex and inhibiting glycogen synthase kinase (GSK)-3β [41]. Myostatin, activin A, and transforming growth factor (TGF)-β, bind activin type 2 receptor B (ActR2B) and induce sarcopenia by activating Smad2/3, which reduces the phosphorylation of Akt [36,40].

Nuclear factor κB (NFκB) is another important molecule regulating signaling pathways associated with skeletal muscle mass maintenance. Activated NFκB increases the expression of MuRF-1 and promotes the ubiquitin-proteasome pathway [42]. NFκB also upregulates proinflammatory cytokines (e.g., TNF-α, IL-1β, and IL-6) and tissue-degrading enzymes (e.g., matrix metalloproteinase-9), which can lead to skeletal muscle depletion [42]. In addition, NFκB reduces MyoD expression and blocks the process of myogenic differentiation [43].

Autophagy is also associated with the maintenance of skeletal muscle mass and function during muscle aging [36]. Age-related muscle dysfunction can be improved by boosting autophagy, which selectively degrades misfolded proteins and dysfunctional organelles [44].

#### 2.3.2. Apoptotic Changes in Myofibers

Apoptosis of myofibers is considered as another mechanism of sarcopenia. Apoptotic loss of skeletal muscle mass can lead to lipid droplet accumulation in skeletal muscle tissues, which is associated with poor skeletal muscle function [45]. Previous studies showed that the activity of caspases, which play central roles in apoptosis, was increased in skeletal muscle tissues of cancer-bearing mouse models [46]. One of the possible mechanisms of apoptotic changes in myofibers is an increased production of reactive oxygen species (ROS). Oxidative stress not only promotes protein degradation, but also causes the fragmentation of mitochondrial DNA and impairs the mitochondrial membrane potential, leading to apoptosis of myofibers [37]. A previous study showed that the mitochondrial oxidative capacities in skeletal muscle tissues decreased in a rat model of cancer cachexia [47], suggesting that cancer cachexia increases oxidative stress in skeletal muscle tissues. Apoptosis of satellite cells in aging skeletal muscle can also induce a loss of skeletal muscle mass and function [48].

### 2.4. Mechanism of Sarcopenia in Cancer Cachexia

In cancer cachexia, various factors, including tumor-derived factors and cytokines, induce sarcopenia due to decreased protein synthesis and increased protein degradation. Moreover, a recent study indicated that increased ROS production, which can be derived from increased lipolysis and excessive fatty acid oxidation, can damage mitochondria in skeletal muscle cells and induce sarcopenia in cancer cachexia [49]. In this section, we summarize the metabolic and molecular backgrounds of sarcopenia associated with cancer cachexia.

#### 2.4.1. Factors Inducing Sarcopenia in Cancer Cachexia

In cancer cachexia, various tumor and host factors influence skeletal muscle mass homeostasis: e.g., proteolysis-inducing factor (PIF) and cytokines, such as TNF-α, and IL-6 [36,37,40]. PIF, a tumor-derived factor detected in the urine of patients with cancer cachexia, can reduce protein synthesis by activating the double-stranded RNA-dependent protein kinase (PKR)/phospholipase A_2_ pathway and can increase protein degradation by promoting nuclear accumulation of NFκB and the ubiquitin-proteasome pathway [37,40].

TNF-α can augment degradation of myofibrillar protein by upregulating Atrogin-1 and MuRF-1 via the inhibition of PI3K/Akt signaling in myofibers [36,40]. In addition, TNF-α upregulates the ubiquitin-proteasome pathway by activating mitogen-activated protein kinase (MAPK) through increased ROS production [50]. Although TNF-α is usually secreted from activated macrophages, it is unclear whether immune cells in adipose or skeletal muscle tissues are associated with increases in the production of TNF-α in cancer cachexia. A recent study showed no benefit of anti-TNF-α antibodies in patients with cancer cachexia [51]. TNF-α is likely to need the synergistic activities of additional tumor-derived factors or cytokines to develop sarcopenia [40].

IL-6 can induce skeletal muscle mass depletion solely and in cooperation with TNF-α. IL-6 is usually host-derived, but it can be secreted from tumor cells, including urothelial cancer cells [52,53]. IL-6 can promote protein degradation by activating both the ubiquitin-proteasome and lysosomal proteolytic pathways in myofibers [37,40].

#### 2.4.2. Oxidative Stress in Sarcopenia Associated with Cancer Cachexia

Oxidative stress is involved in the mechanisms of sarcopenia, including decreased protein synthesis, increased protein degradation, and the upregulation of apoptosis (Figure 2). Oxidative stress upregulates the ubiquitin-proteasome pathway by activating NFκB [54]. Oxidative stress also promotes apoptosis of myofibers by damaging mitochondrial DNA [55]. Recently, Fukawa et al. elucidated the mechanisms of increased ROS production in the mitochondria of skeletal muscle cells [49]. They demonstrated that excessive fatty acid oxidation increases production of ROS and eventually impairs muscle growth by activating MAPK in human muscle stem cell-based models and mouse models with human cancer-induced cachexia [49]. According to their study, fatty acid oxidation is excessively activated in skeletal muscle tissues of cancer cachexia models, whereas glycolysis is downregulated [49], suggesting the role of lipolysis activated by ATGL and HSL in providing skeletal muscle tissues with fatty acids as “fuels for ROS production”. Thus, increased oxidative stress may promote sarcopenia through activated lipolysis and excessive fatty acid oxidation during the course of cancer cachexia. These processes could be therapeutic targets to prevent the progression of sarcopenia and cancer cachexia in cancer-bearing patients.

### 2.5. Sarcopenic Obesity in Cancer

Sarcopenic obesity, which is characterized by low skeletal muscle mass and high adiposity, is considered as the “worst-case scenario” because it includes two adverse conditions: Sarcopenia and obesity. Sarcopenic obesity is related to markedly high morbidity and mortality rates in various diseases [56]. Sarcopenia and obesity generally have several etiologies, including physical inactivity, poor nutrition, inflammation, and insulin resistance. Sarcopenia and obesity can interplay and exacerbate each other [56] (Figure 3). Obesity increases insulin resistance, which promotes skeletal muscle depletion [57]. Moreover, obesity increases fat infiltration in skeletal muscle tissues, leading to poor physical function and performance [45]. Meanwhile, sarcopenia impairs physical activity and function, worsening obesity. Severely obese patients tend to develop sarcopenic obesity as a result of cancer cachexia; although the loss of adipose tissue mechanistically precedes skeletal muscle depletion in cancer cachexia, skeletal muscle depletion progresses in the process of lipolysis of excessive adipose tissues [39].

## 3. Post-Therapeutic Changes in Skeletal Muscle Mass as Prognostic Factors

Recently, several studies demonstrated that changes in skeletal muscle mass are significantly associated with prognosis in cancer-bearing patients [25,26,58]. A decrease in skeletal muscle mass during palliative chemotherapy was significantly associated with poor prognosis in lung cancer patients [59]. Loss of skeletal muscle mass was shown to be a predictor of worse survival in ovarian cancer patients treated with neoadjuvant chemotherapy [60]. A recent study showed that a decrease in skeletal muscle mass during first-line sunitinib therapy was a significant predictor of progression-free survival and OS in metastatic renal cell carcinoma patients [61]. In urothelial carcinoma, the recovery of skeletal muscle mass after chemotherapy reflected a favorable prognosis in advanced urothelial carcinoma patients treated with first-line platinum-based chemotherapy [34]. Moreover, a postoperative reduction in skeletal muscle mass was a significant predictor of shorter OS in bladder cancer patients undergoing radical cystectomy [62]. This clinical evidence indicates the prognostic importance of dynamic changes in skeletal muscle mass during the course of treatment. In this section, we discuss the possible biological mechanisms of changes in skeletal muscle mass during the course of treatment in cancer-bearing patients, especially focusing on skeletal muscle regeneration by the activation of satellite cells.

### 3.1. Possible Mechanisms for the Increase in Skeletal Muscle Mass

We previously reported that skeletal muscle mass recovery after two cycles of chemotherapy was significantly associated with a favorable prognosis in advanced urothelial carcinoma patients treated with first-line platinum-based chemotherapy [34]. In this study, 15 (21%) of the total 72 patients showed post-therapeutic skeletal muscle mass recovery and their prognosis was favorable (two-year OS rate, 69%). Meanwhile, 23 (32%) patients whose skeletal muscle mass had decreased despite a favorable response to systemic chemotherapy showed poor prognosis (two-year OS rate, 19%). This suggests that sarcopenia is reversible in urothelial carcinoma patients and that some interventions that increase skeletal muscle mass may improve the prognosis.

Because skeletal muscle depletion results from hypotrophy of myofibers, post-therapeutic skeletal muscle mass can recover with hypertrophic changes in myofibers. Post-therapeutic alleviation of systemic inflammation due to reduced secretion of cytokines attenuates the catabolic process of cancer cachexia. Moreover, decreased secretion of tumor-derived factors, such as PIF, may underlie the mechanism of hypertrophic changes in myofibers especially in patients who show tumor shrinkage or eradication by systemic chemotherapy, surgical excision, or radiation therapy.

Another possible mechanism of post-therapeutic skeletal muscle mass recovery may be skeletal muscle regeneration. When skeletal muscle tissues are injured, myofibers have no regenerative capacity per se. However, satellite cells can contribute to regeneration of the skeletal muscle [63]. When muscle tissues are injured, satellite cells are activated and differentiated into myoblasts and finally myofibers. Both the quantity and quality of satellite cells are associated with their regenerative capacity [64]. In healthy individuals, aging mostly affects the quality of satellite cells [64]. An in vitro study showed that satellite cells become refractory to the activation by nitric oxide or stretch as age advances [65]. Thus, the recovery of skeletal muscle mass is expected to be better in younger patients. Because oxidative stress can result in a functional loss of satellite cells [66], a reduction in proinflammatory and inflammatory cytokines and oxidative stress by successful cancer treatments would contribute to the recovery of skeletal muscle mass. In addition, activation of NFκB in skeletal muscle tissues can attenuate the regenerative potential of satellite cells [67]. Myostatin also has adverse effects on the proliferative and differentiating potential of satellite cells through the downregulation of paired box 7 (PAX7) [68]. Because obesity and insulin resistance increase the expression of myostatin, changes in skeletal muscle mass may be affected by metabolic syndromes, including diabetes mellitus and sarcopenic obesity.

### 3.2. Possible Therapeutic Interventions to Increase Skeletal Muscle Mass

Conventional treatments for sarcopenia, such as protein supplementation, have limited efficacy for the improvement of sarcopenia in patients with cancer cachexia [69]. As described above, satellite cells, which play an important role in skeletal muscle regeneration, could be a possible therapeutic target. In this section, we discuss potential therapies for sarcopenia, focusing on therapeutic interventions that may activate satellite cells to promote skeletal muscle regeneration.

#### 3.2.1. Anti-Oxidants

Increased oxidative stress not only induces hypotrophic or apoptotic changes in myofibers, but also impairs the regenerative capacity of satellite cells. Thus, anti-oxidants, such as ascorbic acid and carotenoids, could be effective in increasing skeletal muscle mass [70]. n-3 fatty acids, including eicosapentaenoic acid and docosahexaenoic acid, can improve sarcopenia by antagonizing superoxide dismutase [71].

Excessive fatty acid oxidation can cause skeletal muscle depletion by increasing ROS production in myofibers [49]. In addition, increased oxidative stress in skeletal muscle tissues may affect the functions of satellite cells [66]. Etomoxir is an inhibitor of fatty acid oxidation, which reduces ROS in skeletal muscle tissues and may be effective in improving sarcopenia [49].

#### 3.2.2. Activin Type 2 Receptor B (ActR2B) Antagonist

ActR2B, a receptor for activin A, mediates protein degradation in skeletal muscle tissues [36]. The ActR2B antagonist can increase skeletal muscle mass by decreasing protein degradation in myofibers [72]. Because myostatin binds to and activates ActR2B, ActR2B antagonists may promote skeletal muscle mass recovery by improving the regenerative capacity of satellite cells [68].

#### 3.2.3. Treatment for Obesity and Insulin Resistance

Obesity increases insulin resistance, which suppresses muscle glucose uptake and protein anabolism, increases myostatin expression, and eventually progresses to sarcopenia [73]. Thus, treatment for obesity and insulin resistance could improve sarcopenia. In fact, peroxisome proliferator activated receptor γ (PPAR-γ) agonists, including rosiglitazone, troglitazone, and pioglitazone, can increase skeletal muscle mass because of the improvement in insulin sensitivity [74].

Exercise, which includes aerobic exercise and resistance training, ameliorates insulin sensitivity and can be effective in treating sarcopenia [75]. In addition, exercise has anti-tumor potential. Exercise induces the secretion of IL-6 from muscle tissues and mobilizes and redistributes IL-6-sensitive natural killer cells to tumor microenvironments through β-adrenergic signaling [76]. Exercise also inhibits tumor growth by activating the Hippo tumor suppressor pathway through β-adrenergic signaling [77].

## 4. Conclusions

Sarcopenia, the loss of skeletal muscle mass, reflects not only the poor general health status and frailty of patients, but also the possible presence of progressive or advanced cancer and cachexia. Reportedly, sarcopenia is a strong adverse prognostic factor in patients with various cancers, including urothelial carcinoma. Moreover, an increase in skeletal muscle mass during the course of treatment predicts a favorable prognosis. In this article, we reviewed the metabolic and molecular basis of cancer cachexia and sarcopenia. From a mechanistic viewpoint, sarcopenia results from hypotrophic or apoptotic changes in myofibers. The PI3K/Akt signaling plays a key role as a general regulator of skeletal muscle homeostasis, including protein synthesis and degradation, in skeletal muscle tissues. Various factors, including PIF (tumor-derived factor), myostatin (skeletal muscle-derived factor), and cytokines, such as TNF-α and IL-6, can facilitate skeletal muscle depletion by inhibiting the PI3K/Akt signaling in cancer cachexia. In addition, oxidative stress promotes skeletal muscle depletion by increasing protein degradation and inducing apoptosis of myofibers by damaging mitochondrial DNA. In cancer cachexia, increased lipolysis and excessive fatty acid oxidation in skeletal muscle tissues causes oxidative stress in myofibers, resulting in sarcopenia. We also discussed possible mechanisms of increasing skeletal muscle mass during the course of treatment. Increases in skeletal muscle mass can result from hypertrophic changes in myofibers or skeletal muscle regeneration via the activation of satellite cells. The alleviation of systemic inflammation or reduced secretion of PIF as a result of cancer treatment may be underlying mechanisms of hypertrophic changes in myofibers. The regenerative capacity of satellite cells is age-dependent and so the recovery of skeletal muscle mass is expected to be better in younger patients. The regenerative potential of satellite cells is regulated by various factors, including oxidative stress. Anti-oxidants may be a promising therapeutic intervention to improve sarcopenia.

Taken together, various metabolic and inflammatory factors and molecular pathways are involved in the advent of sarcopenia and the increase in skeletal muscle mass during the course of treatment. The multifactorial nature of sarcopenia, involving general health status, nutritional status, systemic inflammation, metabolic abnormality, oxidative stress, and cancer progression, may support its usefulness as a biomarker in the management of urothelial carcinoma patients, especially in predicting prognosis.

## Figures and Tables

**Figure 1 ijms-20-00760-f001:**
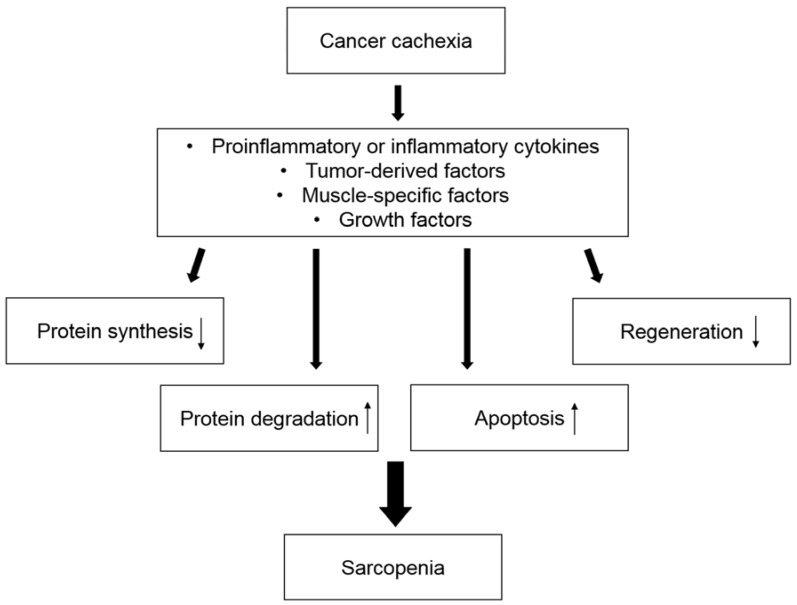
An outline of sarcopenia development in cancer cachexia. Sarcopenia is caused by decreased protein synthesis, increased protein degradation, upregulated apoptosis, and downregulated skeletal muscle regeneration. Various mediators are involved in these processes.

**Figure 2 ijms-20-00760-f002:**
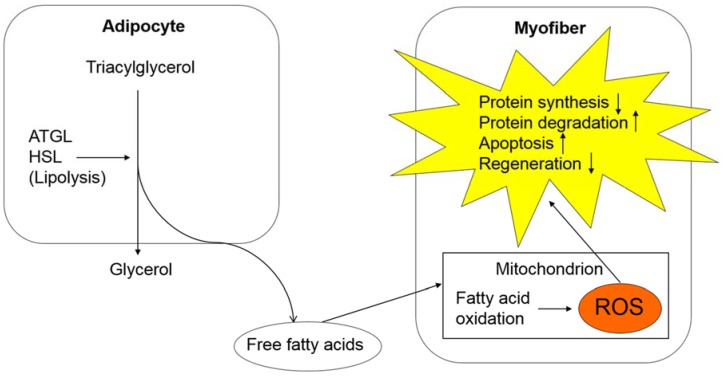
Oxidative stress in cancer cachexia and its effect on skeletal muscle tissues. In adipose tissues, lipolysis is upregulated, which results in increased fatty acid production. A large amount of fatty acids undergo an oxidation process in the mitochondria of myofibers, causing an excessive production of ROS. ROS is involved in various processes, including decreased protein synthesis, increased protein degradation, upregulated apoptosis, and downregulated skeletal muscle regeneration. Abbreviations: Adipose triglyceride lipase, ATGL; hormone-sensitive lipase, HSL; reactive oxygen species, ROS.

**Figure 3 ijms-20-00760-f003:**
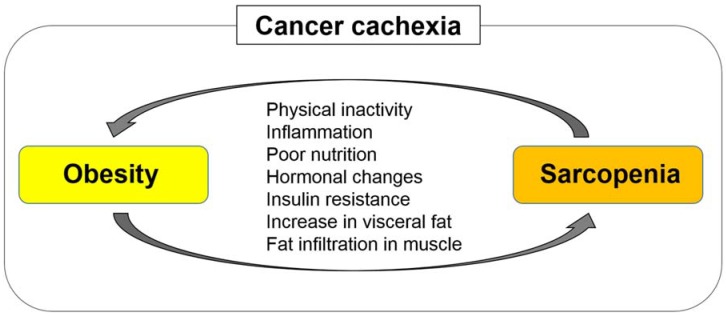
Sarcopenia and obesity can interplay and exacerbate each other. Sarcopenia and obesity have common etiologies, including physical inactivity, inflammation, poor nutrition, hormonal changes, and insulin resistance. Obesity can facilitate skeletal muscle depletion and fat infiltration in skeletal muscle tissues through increased secretion of inflammatory cytokines and increased insulin resistance, leading to poor physical function and performance. Meanwhile, sarcopenia impairs physical activity and function, increasing visceral fat and worsening obesity.

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
