# Peer review of "Metabolic and Molecular Basis of Sarcopenia: Implications in the Management of Urothelial Carcinoma"

_ijms, 2019, doi:10.3390/ijms20030760_

Round 1

Reviewer 1 Report

The manuscript titled ‘Metabolic and molecular basis of sarcopenia----‘ by Fukushima et al. reviews metabolic and molecular basis of sarcopenia in cancer patients and suggests various management measures to improve cancer-associated sarcopenic condition. The manuscript introduced molecular aspects of sarcopenia and cancer cachexia in relatively reasonable way. Although the authors described mechanism of sarcopenia and cancer cachexia superficially, they provided an overview on the phenomena wide enough to bring about interest in the area. Since the title reflects urothelial cancer, the authors are recommended to provide more detailed and focused description on the phenomena specifically pertinent to urothelial cancer.

Author Response

First of all, we greatly appreciate the reviewers for taking their precious time to review our paper and giving constructive comments. We revised the manuscript according to the reviewers’ comments. Please take a look at the changes we made in the revised version outlined below.

The manuscript titled ‘Metabolic and molecular basis of sarcopenia----‘ by Fukushima et al. reviews metabolic and molecular basis of sarcopenia in cancer patients and suggests various management measures to improve cancer-associated sarcopenic condition. The manuscript introduced molecular aspects of sarcopenia and cancer cachexia in relatively reasonable way. Although the authors described mechanism of sarcopenia and cancer cachexia superficially, they provided an overview on the phenomena wide enough to bring about interest in the area. Since the title reflects urothelial cancer, the authors are recommended to provide more detailed and focused description on the phenomena specifically pertinent to urothelial cancer.

I appreciate the reviewer’s constructive comment. The prognostic role of sarcopenia has been reported in various cancers. In particular, many studies reported that sarcopenia was significantly associated with worse prognosis in patients with urothelial carcinoma, including bladder cancer, upper tract urothelial carcinoma, and advanced urothelial carcinoma. These findings suggest that sarcopenia can be clinically relevant especially for the management of patients with urothelial carcinoma. To clarify this point, we added the following sentence to the section 1:

“These findings suggest that sarcopenia can be a clinically relevant syndrome especially in urothelial carcinoma.” (Page 2, line 86-87)

Reviewer 2 Report

In this manuscript, the authors have discussed the molecular mechanisms and diagnostic significance of skeletal muscle mass loss in bladder cancer. The authors described in detail how signaling mechanisms affect protein synthesis and degradation in the skeletal muscles. They also described the possible effects of lipid metabolism in adipocytes on skeletal muscle mass. They further discussed possible approaches on prognosis and treatment of bladder cancer with the aid of knowledge about sarcopenia. This review covers an important topic of cancer therapy and is logically constructed. Therefore, this would be helpful for readers to understand the basic concepts around this topic.

Minor point,

L. 255    In section 3, the term "biomarker" is used to stand for the significance of muscle atrophy in cancer diagnosis. To my understanding, however, a biomarker is typically a molecule that are increased or decreased in the blood, urine or tissue samples reflecting diagnostic or prognostic situations. Once a possible biomarker candidate is found in an explorative research, its diagnostic value is evaluated by further experiments with patients and healthy controls, whose numbers are enough large to obtain a conclusion by statistical analysis. So it looks like muscle atrophy is not a typical "biomarker". I wonder if there may be some other expressions that would fit better to the content.

Author Response

First of all, we greatly appreciate the reviewers for taking their precious time to review our paper and giving constructive comments. We revised the manuscript according to the reviewers’ comments. Please take a look at the changes we made in the revised version outlined below.

In this manuscript, the authors have discussed the molecular mechanisms and diagnostic significance of skeletal muscle mass loss in bladder cancer. The authors described in detail how signaling mechanisms affect protein synthesis and degradation in the skeletal muscles. They also described the possible effects of lipid metabolism in adipocytes on skeletal muscle mass. They further discussed possible approaches on prognosis and treatment of bladder cancer with the aid of knowledge about sarcopenia. This review covers an important topic of cancer therapy and is logically constructed. Therefore, this would be helpful for readers to understand the basic concepts around this topic.

Minor point,

L. 255    In section 3, the term "biomarker" is used to stand for the significance of muscle atrophy in cancer diagnosis. To my understanding, however, a biomarker is typically a molecule that are increased or decreased in the blood, urine or tissue samples reflecting diagnostic or prognostic situations. Once a possible biomarker candidate is found in an explorative research, its diagnostic value is evaluated by further experiments with patients and healthy controls, whose numbers are enough large to obtain a conclusion by statistical analysis. So it looks like muscle atrophy is not a typical "biomarker". I wonder if there may be some other expressions that would fit better to the content.

I appreciate the reviewer’s constructive comment. We changed the term “biomarker” into “factor” in this sentence. Thus, we revised the sentence as described below:

“3. Post-therapeutic changes in skeletal muscle mass as prognostic factors” (Page 7, line 256)